

# Function and regulation of plant ARGONAUTE proteins in response to environmental challenges: a review

Uroosa Zaheer[1,2,3], Faisal Munir[1,2,3], Yussuf Mohamed Salum[1,2,3] and Weiyi He[1,2,3]

[1] Plant Protection, State Key Laboratory for Ecological Pest Control of Fujian and Taiwan Crops, Institute of Applied Ecology, Fujian Agriculture and Forestry University, Fuzhou, Fujian, China
[2] Plant Protection, International Joint Research Laboratory of Ecological Pest Control, Ministry of Education, Fujian Agriculture and Forestry University, Fuzhou, Fujian, China
[3] Plant Protection, Ministerial and Provincial Joint Innovation Centre for Safety Production of Cross-Strait Crops, Fujian Agriculture and Forestry University, Fuzhou, Fujian, China

## ABSTRACT

Environmental stresses diversely affect multiple processes related to the growth, development, and yield of many crops worldwide. In response, plants have developed numerous sophisticated defense mechanisms at the cellular and subcellular levels to react and adapt to biotic and abiotic stressors. RNA silencing, which is an innate immune mechanism, mediates sequence-specific gene expression regulation in higher eukaryotes. ARGONAUTE (AGO) proteins are essential components of the RNA-induced silencing complex (RISC). They bind to small noncoding RNAs (sRNAs) and target complementary RNAs, causing translational repression or triggering endonucleolytic cleavage pathways. In this review, we aim to illustrate the recently published molecular functions, regulatory mechanisms, and biological roles of AGO family proteins in model plants and cash crops, especially in the defense against diverse biotic and abiotic stresses, which could be helpful in crop improvement and stress tolerance in various plants.

## INTRODUCTION

The emerging extreme weather caused by drastic and rapid climate change hampers plant development and growth, threatening crop productivity worldwide (*Farooq et al., 2022*; *Singh et al., 2023*). This includes a range of stresses, both biotic (bacteria, fungi, insects, nematodes, and viruses) and abiotic (extreme temperatures, salinity, heavy metals, UV-B radiation, and drought), and the broader influence of climate change on environmental conditions, all of which contribute to adverse impacts on plant responses (*Raza et al., 2019*; *Zandalinas, Fritschi & Mittler, 2021*). These stresses may sequentially or simultaneously affect plant yield by diminishing several physiological, molecular, biochemical, and morphological processes (*Zandalinas & Mittler, 2022*). Plants activate complex regulatory mechanisms such as altered signal transduction and gene regulation networks at the transcriptional or translational level to cope with environmental stresses (*Zhang et al.,*

Corresponding author
Weiyi He, wy.he@fafu.edu.cn

*2022a*; *Zhang et al., 2022b*). RNA interference (RNAi) is a conserved mechanism in various organisms that plays vital roles in multiple processes, including growth, development, defenses, and stress responses against abiotic and biotic stresses (*Muhammad et al., 2019*; *Zhu & Palli, 2020*; *Qiao et al., 2021*). Three protein complexes—Dicer-like (DCL), ARGONAUTE (AGO), RNA-dependent RNA polymerase (RDR)—are essential for the RNA silencing process (*Cui et al., 2020*; *Li et al., 2023*).

AGOs are core elements of the RNA-induced silencing complex (RISC) that participate in the loading of small RNAs (sRNAs) (*Liao et al., 2020*; *Moussian & Casadei, 2022*; *Yun & Zhang, 2023*). AGOs bind to sRNA complexes, including small interfering RNAs (siRNAs) or microRNAs (miRNAs), and subsequently target and silence complementary RNA or DNA through the mechanism of transcriptional gene silencing (TGS) or posttranscriptional gene silencing (PTGS) (*Carbonell, 2017*; *Yang & Li, 2018*; *Zhao et al., 2023*). Complementary RNA sequences are silenced through independent cleavage mechanisms such as direct endonucleolytic cleavage or translational repression and target destabilization (*Guo, Li & Ding, 2019*; *Feng et al., 2021*). TGS occurs through chromatin remodeling or RNA-dependent DNA methylation to regulate target genes (*Matzke, Kanno & Matzke, 2015*). In addition, RNA silencing integrates with autophagy (*Li et al., 2017a*; *Li et al., 2017b*), ubiquitination (*Cheng & Wang, 2017*), and R gene-mediated immunity (*Zhu et al., 2019*) in plants for defense responses.

AGOs have been identified in numerous plants and play essential roles in the response to various biotic and abiotic stresses (*Fang & Qi, 2016*; *Liu et al., 2021*). Here, we summarize recently published molecular and biological functions of AGO protein families in plants in response to diverse environmental stresses, which may provide insights into RNAi-based molecular approaches for researchers working on the genetic improvement of the stress tolerance of major economic crops.

### Survey methodology

Literature relevant to the subject of this manuscript was evaluated using Google Scholar, PubMed databases, and appropriate academic journals. A combination of keywords was used to search for relevant terms in the literature, such as "ARGONAUTE proteins", "small RNAs", "endonucleolytic cleavage", "translational repression", "DNA methylation", "gene regulation", "biotic stresses", and "abiotic stresses". All the articles were read in detail, and their subject matter was evaluated. Articles were included if they provided insights into the molecular functions, regulatory mechanisms, and biological roles of AGO proteins in combating environmental stresses in plants.

## CLASSIFICATION OF PLANT AGOS

AGOs have been identified from bacteria, archaea, and eukaryotes (*Olina et al., 2018*). All AGOs are evolutionarily conserved in structure and function in cellular development. These proteins are involved in alternative splicing and host defense (*Singh et al., 2015*; *Yang, Cho & Zheng, 2020*). However, the number of AGO proteins varies from a single protein in the yeast *Schizosaccharomyces pombe* to 27 in *Caenorhabditis elegans* (*Swarts et al., 2014*). In plants, the number of genes encoding AGO proteins ranges from seven in *Cucumis*

*sativus* (*Gan et al., 2017*) to 69 in *Triticum aestivum* (*Liu et al., 2021*) (Table 1). AGOs have a high molecular weight of approximately 100 kDa (*Mirzaei et al., 2014*), and these proteins possess four functional domains, a variable N domain, and three conserved domains (PIWI, MID, and PAZ) (Fig. 1) (*Bologna & Voinnet, 2014*; *Carbonell, 2017*). These domains have different functions and features for loading sRNAs and RISC activity (*Ipsaro & Joshua-Tor, 2015*). Among them, the N domain engages in the unwinding of sRNA duplexes. The PAZ and MID domains bind the 3′ nucleotide and 5′ monophosphate nucleotide of sRNA, respectively. The PIWI domain associates with siRNA at the 5′ end and functions as a ribonucleolytic domain containing four metal-coordinating residues (aspartate-aspartate-histidine-histidine) that are crucial for slicing activity (*Chen et al., 2021*; *Liu et al., 2021*). Two additional domains, Argo-L1 and Argo-L2, are present alongside these domains in AGO proteins (*Krishnatreya et al., 2021*). Plant AGOs interact with sRNAs *via* several structural features, including the 5′ nucleotide, PIWI domain, and duplex structure (*Zhang et al., 2014*).

AGO proteins have diverged into three paralogous groups (Fig. 2). The first group includes AGO-like proteins, accounting for most AGO proteins that are homologous to *Arabidopsis thaliana* AGO1; the second group comprises PIWI-like proteins similar to *Drosophila melanogaster* PIWI and expressed in animal germ cells; and the third group comprises AGOs expressed only in *C. elegans* (*Bajczyk et al., 2019*). AGO1 and AGO2 participate in pathways involving miRNAs, siRNAs, trans-acting siRNAs (tasiRNAs), PTGS, and antiviral agents in different plants (*Niedojadło et al., 2020*). Furthermore, AGO proteins in flowering plants can be classified into three major classes according to phylogenetic relationships: AGO1/5/10, AGO2/3/7, and AGO4/6/8/9 (*Rodríguez-Leal et al., 2016*; *Li et al., 2021*). Notably, monocot plants, including rice, maize, and sugarcane, have evolved an AGO18 group as part of the AGO1/5/10 class (*Cui et al., 2020*).

# MOLECULAR FUNCTION OF PLANT AGOS

The molecular mode of action of plant AGOs is described in Fig. 3.

## Endonucleolytic cleavage

AGOs bind to various sRNAs and subsequently silence transposable elements or target genes at the transcriptional or posttranscriptional level (*Fang & Qi, 2016*; *Ré et al., 2020*). The PIWI domain contains a metal-coordinating catalytic tetrad of Asp-Glu-Asp-His amino acids and cleaves the target complementary RNA by utilizing RNase H-like activity (*Choudhary et al., 2021*). AGOs associate with miRNAs and cleave target mRNAs at the 10th and 11th nucleotides from the 5′ end of the miRNA, thus regulating gene expression (*Pegler, Grof & Eamens, 2019*). In *A. thaliana*, various studies confirmed the slicer activity of AGO proteins by catalyzing the endonucleolytic cleavage of target RNAs such as AtAGO1, AtAGO2, AtAGO4, AtAGO7, and AtAGO10 (*Schuck et al., 2013*; *Carbonell, 2017*). Among AtAGOs, AtAGO1 is the best-characterized member in plants; it binds to miRNAs or siRNAs from endogenous loci or viruses, cleaves target RNAs, and represses translation activity (*Borges & Martienssen, 2015*; *Ma & Zhang, 2018*). AtAGO10 is involved in the degradation of miR165/6 by sRNA-degrading nucleases (SDNs) and plays an essential role in stem

**Table 1  Numbers of AGOs in different plants.**

| Plant Species | Type of organism | Number | Reference |
|---|---|---|---|
| *Oryza sativa* (Rice) | Monocot Plant | 19 | *Kapoor et al. (2008)* |
| *Zea mays* (maize) | Monocot Plant | 18 | *Qian et al. (2011)* |
| *Sorghum bicolor* (Sorghum) | Monocot Plant | 16 | *Liu et al. (2014)* |
| *Setaria italica* (Foxtail millet) | Monocot Plant | 19 | *Yadav et al. (2015)* |
| *Brachypodium distachyon* (False brome) | Monocot Plant | 16 | *Šečić, Zanini & Kogel (2019)* |
| *Saccharum spontaneum* (Sugarcane) | Monocot Plant | 21 | *Cui et al. (2020)* |
| *Hordeum vulgare L* (Barley) | Monocot Plant | 11 | *Hamar et al. (2020)* |
| *Triticum aestivum* (Bread wheat) | Monocot Plant | 69 | *Liu et al. (2021)* |
| Musa acuminate (Banana) | Monocot Plant | 13 | *Ahmed et al. (2021)* |
| *Arabidopsis thaliana* (Arabidopsis) | Dicot Plant | 10 | *Vaucheret (2008)* |
| *Solanum lycopersicum* (Tomato) | Dicot Plant | 15 | *Xian et al. (2013)* |
| *Glycine max* (Soybean) | Dicot Plant | 21 | *Liu et al. (2014)* |
| *Vitis vinifera* (Grapevine) | Dicot Plant | 13 | *Zhao et al. (2015b)* |
| *Populus trichocarpa* (Black cottonwood) | Dicot Plant | 15 | *Zhao et al. (2015b)* |
| *Phaseolus vulgaris* (Common bean) | Dicot Plant | 17 | *De Sousa Cardoso et al. (2016)* |
| *Malus domestica* (Apple) | Dicot Plant | 15 | *Xu et al. (2016a)* and *Xu et al. (2016b)* |
| *Brassica napus* (Rapeseed) | Dicot Plant | 27 | *Cao et al. (2016a)* |
| *Brassica rapa* (Turnip mustard) | Dicot Plant | 13 | *Cao et al. (2016a)* |
| *Brassica oleracea* (Wild cabbage) | Dicot Plant | 14 | *Cao et al. (2016b)* |
| *Cucumis sativus* (Cucumber) | Dicot Plant | 7 | *Gan et al. (2017)* |
| *Coffea* (Coffee) | Dicot Plant | 11 | *Noronha Fernandes-Brum et al. (2017)* |
| *Cicer arietinum* (Chickpea) | Dicot Plant | 13 | *Garg et al. (2017)* |
| *Cajanus cajan* (Pigeon pea) | Dicot Plant | 13 | *Garg et al. (2017)* |
| *Arachis ipaensis* (Peanut) | Dicot Plant | 11 | *Garg et al. (2017)* |
| *Capsicum annum* (Pepper) | Dicot Plant | 12 | *Qin et al. (2018)* |
| *Citrus sinensis* (Sweet orange) | Dicot Plant | 13 | *Sabbione et al. (2019)* |
| *Solanum tuberosum* (Potato) | Dicot Plant | 14 | *Liao et al. (2020)* |
| *Camellia sinensis* (Tea) | Dicot Plant | 18 | *Krishnatreya et al. (2021)* |
| *Dimocarpus longan lour* (Longan) | Dicot Plant | 10 | *Chen et al. (2021)* |
| *Gossypium arboretum* (Tree cotton) | Dicot Plant | 14 | *Fu et al. (2022)* |
| *Gossypium hirsutum* (Mexican cotton) | Dicot Plant | 28 | *Fu et al. (2022)* |
| *Fragaria vesca* (Woodland strawberry) | Dicot Plant | 13 | *Jing et al. (2023)* |
| *Fragaria x ananassa* (Strawberry) | Dicot Plant | 33 | *Jing et al. (2023)* |

cell maintenance (*Yu et al., 2017*). The slicing of AGO2, AGO7, and miRNA-associated AGO1 is crucial for antiviral activity, young-to-adult phase transformation, and plant growth, respectively (*Carbonell et al., 2012*). Moreover, AGO2 and AGO1 have been shown to function in silencing pseudogenes, transposons, and intergenic regions (*Garcia et al., 2012*). The slicing of plant AGOs is essential for amplifying phasiRNAs (phased small interfering RNAs) or tasiRNAs from target transcripts (*Rogers & Chen, 2013*). Slicing by AGO1-associated miRNAs also occurs on membrane-bound polysomes, thus enabling the production of phasiRNAs from membrane-bound noncoding precursors in *A. thaliana* (*Li*

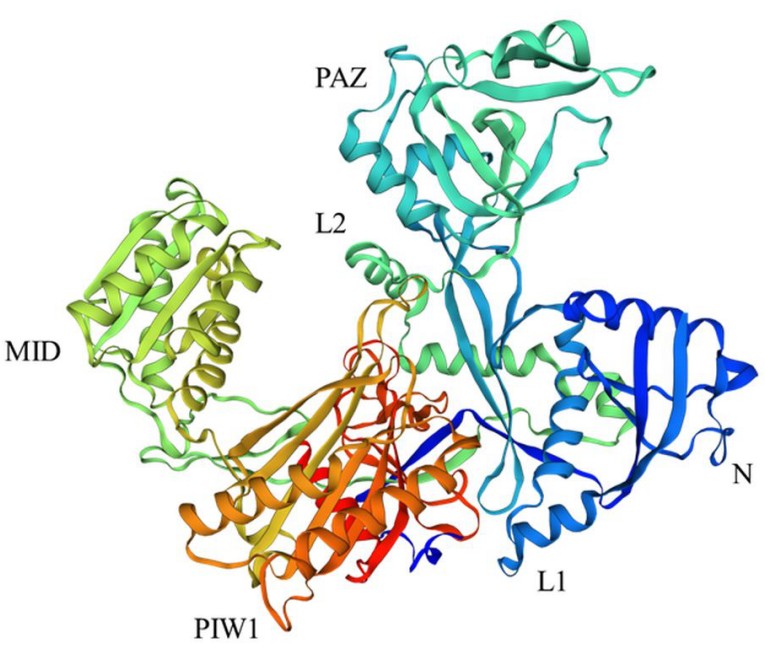

**Figure 1 Structure of AGO18 protein from *Saccharum spontaneum*.** Crystal structure of plant AGOs: AGO consist of a variable N-domain, three conserved domains PIWI, MID, and PAZ, and two linker domains Argo-L1 and Argo-L2. N domain was presented by dark blue color, PAZ by sky blue, MID by lime, PIWI by orange, L1 by blue, and L2 by light green. The structure was constructed by the SWISS Model (http://swissmodel.expasy.org) online server using the AGO18 sequence from *Saccharum spontaneum*.

*et al., 2016*). The slicer activity of AGO1 is required specifically for the phasing of but not the production of tasiRNA in *A. thaliana* (*Arribas-Hernández et al., 2016*).

Antiviral RNA silencing, including the use of viral siRNAs (vsiRNAs), has been associated with RISC complexes that contain AGO nucleases and other characterized components. The antiviral gene AGO1 binds to vsiRNAs and cleaves viral RNAs. The cleaved RNAs are used as substrates for DCL2, SGS3, and RDR6 to produce secondary vsiRNAs. The secondary vsiRNAs are then loaded onto the antiviral AGO2 to treat viral RNA infection. In addition, AGO1, AGO2, AGO3, AGO7, and AGO10 mediate antiviral resistance by associating with viral siRNAs (*Garcia-Ruiz et al., 2015*; *Fang & Qi, 2016*). AGOs associate with sRNAs and regulate gene expression by catalyzing the endonucleolytic cleavage of target RNAs in plants. Slicing by AGOs is essential for plant development and antiviral stress resistance.

## Translational repression

Plant miRNAs are associated with AGOs and form a miRNA-induced silencing complex (miRISC) to silence mRNA expression (*Muhammad et al., 2019*; *Jungers & Djuranovic, 2022*). The translation repression mediated by AGO1-miRNA depends on the variable positions of the miRNA sites. AGO1-mediated miRISC targets 5′ untranslated regions, which can block translation initiation and ribosome recruitment. Conversely, the AGO1-associated RISC targeting the ORF blocks translation elongation and ribosome movement.
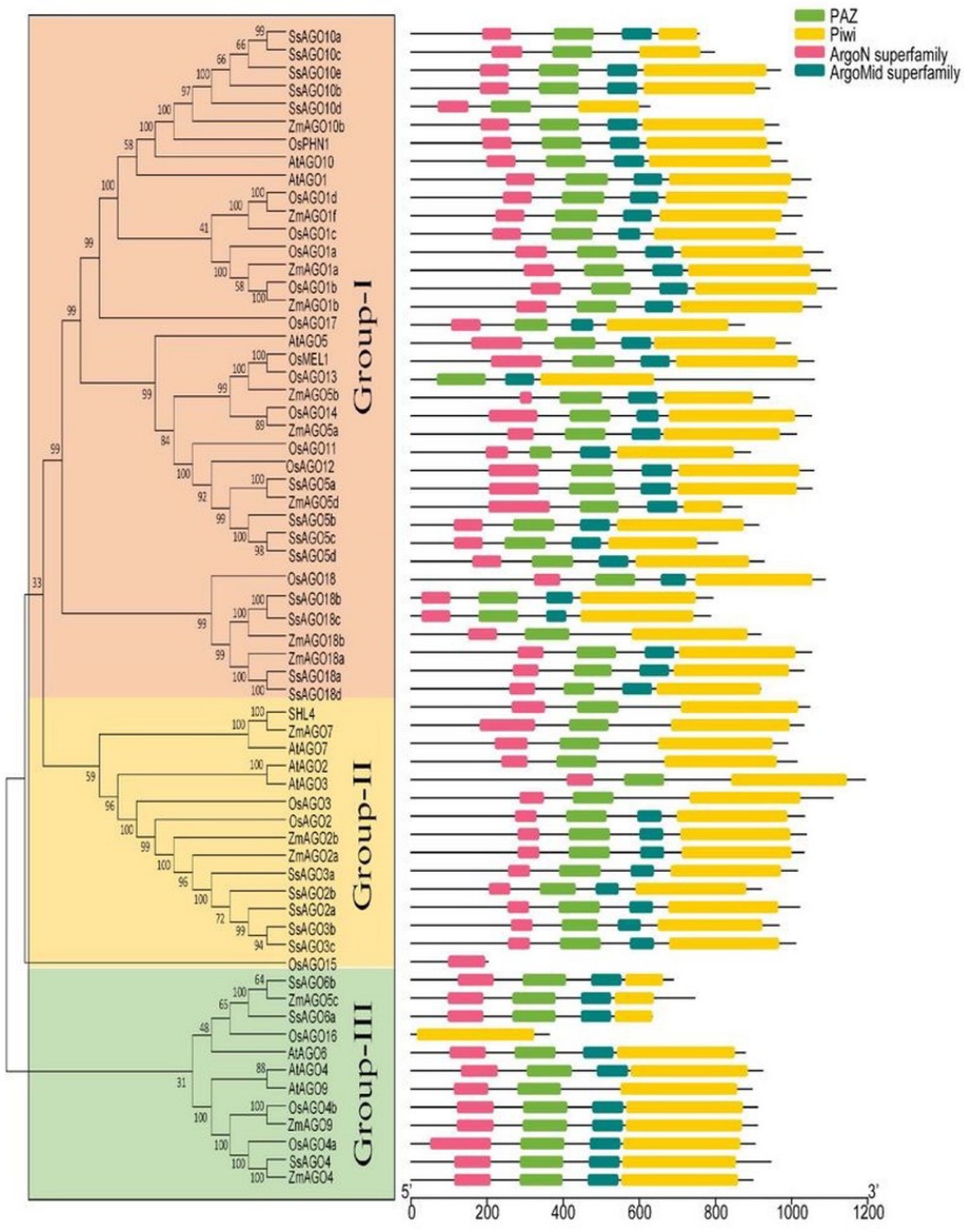

**Figure 2 Phylogenetic analysis of AGOs.** The classification and representative domains analysis of different AGOs groups: Mega 6 software was used for the construction of phylogenetic tree. Bootstrap support values from 1,000 replications were indicated above the branches. The I-III roman numerals showed different AGO's groups. The conserved domains were identified with MEME (http://meme-suite.org/index.html).

Plant AGOs associated with miRNAs bind to the 3′UTR and prompt translational repression (*Iwakawa & Tomari, 2013*; *Merchante, Stepanova & Alonso, 2017*). In *A. thaliana*, the AtAGO1 protein associates with miRNAs in the cytoplasm and mediates translational

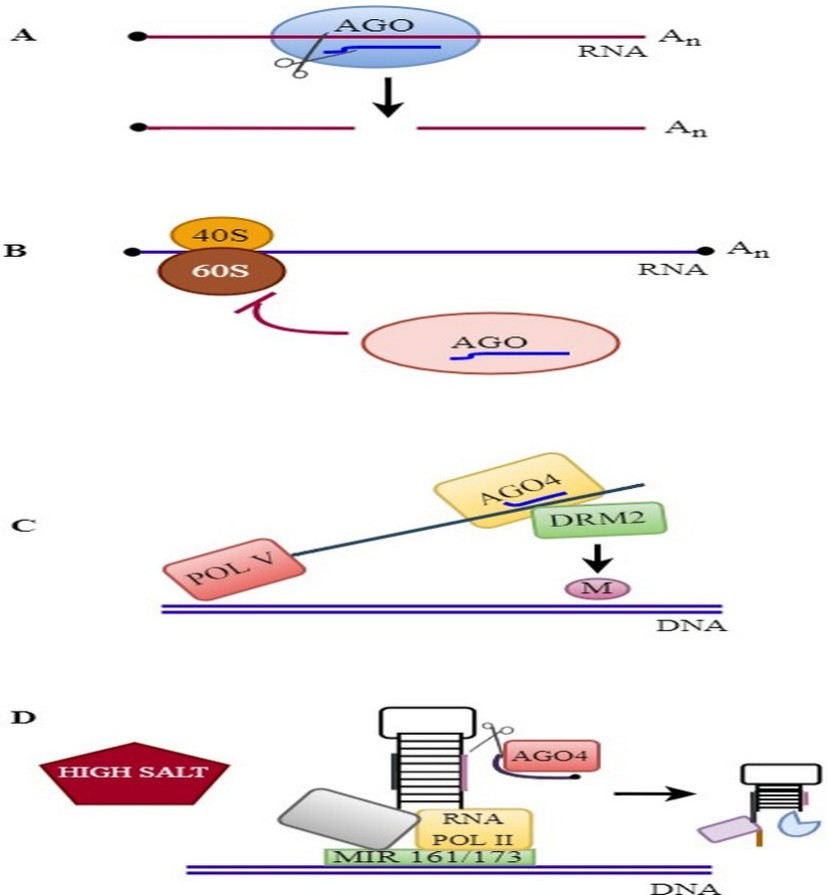

**Figure 3  RNAi mechanisms in plants.** RNAi mechanisms in plants: (A) Endonucleolytic cleavage: AGOs bind to various sRNAs and silence the target genes. (B) Translational repression: AGOs associates with miRNA and suppress the translation. (C) RdDM pathway: AGO4 binds to siRNA and mediates DNA methylation. (D) MiRNA gene regulation: AGO4 regulates the slicing of *MIR161/MIR173* precursors and controls miRNA production against salinity stress.

repression at the posttranscriptional level to suppress target genes (*Liu et al., 2018a*; *Liu et al., 2018b*). AtAGO10 has been associated with miR168 and demonstrated to suppress translation efficiency in *A. thaliana* (*Pumplin & Voinnet, 2013*). Furthermore, AGO2-associated miRISC targets the ORF or 3′UTR and appears to exert translational repression and antiviral functions in plants (*Fátyol, Ludman & Burgyán, 2016*). AGO-associated miRNAs have been implicated to function with both monosomes and polysomes at the translational level (*Li et al., 2016*). Translational repression of AGO1-miRNAs occurs in the endoplasmic reticulum, and the ALTERED MERISTEM PROGRAM 1 (AMP1) membrane protein is involved in the translational repression of target miRNAs (*Li et al., 2013*). Additionally, it was observed that Hyponastic Leaves 1 (HYL1) binds to AMP1 and AGO1 and promotes translational suppression (*Yang et al., 2021*).

The ability of AGO1 to repress translation is associated with the efficient biogenesis of 22-nt siRNAs, which mediate stress adaptation (*Wu et al., 2020*). Antiviral AGOs bind to
viral sRNAs and suppress complementary DNAs or RNAs, thus promoting plant antiviral stress resistance. Moreover, the severity of infection caused by viruses has been associated with the efficacy of viral suppressors of RNA silencing (VSRs) (*Ramesh et al., 2021*). It was demonstrated that AGO18 has a significant role in slicing activity, binds to miR168, and alleviates the suppression of AGO1. This association increases rice AGO1 accumulation and antiviral defense (*Wu et al., 2015*). The suppressor protein AC2 of mungbean yellow mosaic India virus interacts with AGO1 and RDR6 to prevent slicing activity (*Kumar et al., 2015*). Many viruses, such as cucumber mosaic virus, tobacco mosaic virus, cymbidium ringspot virus, and turnip crinkle virus, induce miR168 expression and suppress posttranscriptional gene silencing, thus activating AGO1 translational suppression (*Li et al., 2017a*; *Li et al., 2017b*). In addition, AGO7 associates with the noncleavable site of miR390, leading to ribosome stacking and thus preventing translational elongation in *A. thaliana* (*Hou et al., 2016*). Similarly, AGO10 modulates meristem development by mediating translational repression of target miRNA genes (*Fang & Qi, 2016*). AGOs bind to miRNAs and repress target genes at the posttranscriptional level by translational repression in plants.

## RNA-dependent DNA methylation

DNA methylation blocks transposon activity, maintains genome stability, controls gene expression, and is crucial for transcriptome modification under abiotic and biotic stress responses (*Zhang, Lang & Zhu, 2018*). In plants, three processes are related to DNA methylation: *de novo* methylation, demethylation, and methylation maintenance pathways (*Kumar & Mohapatra, 2021*). *De novo* methylation is mediated by the binding of AGO4 to 24-nt siRNAs and is essential for RNA-dependent DNA methylation (RdDM) pathways (*Mbichi, Wang & Wan, 2020*). RdDM pathways are facilitated by the SAWADEE HOMEODOMAIN HOMOLOG 1/DNA BINDING TRANSCRIPTION FACTOR1 (SHH1/DBTF1) protein and four chromatin remodeling proteins, CLSY1 to CLSY4 (*Zhou, Palanca & Law, 2018*). It is initiated by the action of RDR2 and RNA polymerase IV to synthesize dsRNAs. dsRNA is cleaved in the nucleus by the DCL3 protein into 24-nucleotide siRNAs, which are then exported to the cytoplasm (*Wendte & Pikaard, 2017*; *Singh et al., 2019*). The binding of AGO4 to 24-nt siRNA targets loci through base pairing with RNA polymerase V (Pol V) transcripts or through binding to glycine-tryptophan (GW) and tryptophan-glycine (WG) motifs (*Lahmy et al., 2016*; *Wang & Axtell, 2017*). AGO4-associated siRNA further recruits the domain-rearranged methyltransferase 2 (DRM2) protein to target cytosines for DNA methylation (*Liu et al., 2018a*; *Liu et al., 2018b*). Furthermore, RDR2, Pol IV, and Pol V are essential components of the RdDM pathway in rice and other plants. These components play a crucial role in mediating the RdDM pathway, contributing to epigenetic regulation in diverse plants (*Wang et al., 2020*).

The RdDM pathway leads to the transcriptional suppression of a subset of genes and transposons and is involved in plant development, stress responses, reproductive mechanisms, genomic imprinting, and pathogen defense (*Matzke & Mosher, 2014*; *Erdmann & Picard, 2020*). They are involved in initiating cytosine DNA methylation at 21- or 24-nt noncoding RNA target sites and confer tolerance against DNA viruses in plants (*Guo, Li & Ding, 2019*). Moreover, studies of the nuclear functions of *A. thaliana*

have indicated that AGO1 is loaded with miRNAs in the nucleus and that AGO1: miRNA complexes associate with chromatin and affect gene transcription (*Bajczyk et al., 2019*). AGO4 or RdDM is involved in regulating the expression of stress-responsive genes *via* chromatin modification (*Au, Dennis & Wang, 2017*). In addition to AGO4, other proteins, including Pol IV, POLV, RDR2, and histone modification proteins in the RdDM pathway, are essential for mediating resistance against viral infection (*Corrêa et al., 2020*). Furthermore, AGO6 and AGO9 are associated with 24-nucleotide siRNAs, mediating the RdDM pathway in *A. thaliana*. AGO6 preferentially binds with heterochromatic siRNAs and participates in the TGS and RdDM pathways. It has been shown that AGO6 can work with AGO4 to promote DNA methylation at target loci, and both of these proteins are dependent on each other (*Duan et al., 2015*). Overall, RdDM is mediated by the binding of AGO4 or AGO6 to 24-nt siRNAs in plants and is involved in plant genome stability.

## miRNA gene regulation

Plant miRNAs are small noncoding RNAs that are 21–24 nt in length and posttranscriptionally reduce gene expression (*Wang, Mei & Ren, 2019*; *Gao, Nie & Wang, 2021*). They participate in a wide range of biological processes, including growth, development, and responses to biotic and environmental stresses, through influencing genomic integrity and regulating metabolism (*Samynathan et al., 2023*). RNA polymerase II transcribes miRNA genes in the nucleus to produce pre-miRNAs. The pre-miRNAs contain long hairpin structures approximately 70 nucleotides in length. Subsequently, a series of processes lead to the formation of a single hairpin structure cleaved by DCL to produce a mature miRNA. Mechanistically, DCL1 is initially associated with the chromatin of the MIR gene (*Fang et al., 2015*). Several regulatory proteins that regulate miRNA transcription, including CDC5, ELP2, and NOT2, have been shown to be associated with processing machinery proteins (*Wang et al., 2019*). Afterward, miRNA is transported to the cytoplasm by HASTY and associated with AGOs to stimulate the degradation of the target transcript through translational repression or endonucleolytic cleavage mechanisms. Plant miRNAs exhibit a high degree of complementarity with their mRNA targets, resulting in the silencing of target mRNAs (*Kar & Raichaudhuri, 2021*; *Begum, 2022*). Furthermore, AGOs play a vital role in the maintenance of miRNA stability. Different proteins, including ATRM2, HESO1, SDN1, and URT1, interact with AGO1 and are implicated in the regulation of AGO1-associated miRNAs and unmethylated miRNAs or miRNA\* duplexes (*Chen et al., 2018*; *Wang et al., 2018*). Similarly, several AGOs are regulated by multiple miRNAs. For example, AGO1 and AGO2 are regulated by miR168 and miR403, respectively, and these regulatory modules control yield-associated traits in rapeseed (Brassica napus) (*Wang et al., 2021a*; *Wang et al., 2021b*; *Zhang et al., 2023*). Moreover, miRNAs also indirectly regulate gene expression by producing secondary siRNAs that are tasiRNAs or phasiRNAs, such as miR828, miR390, and miR173 (*De Felippes, 2019*; *Zhang et al., 2022a*; *Zhang et al., 2022b*), which are involved in several developmental and physiological pathways (*Li et al., 2017a*; *Li et al., 2017b*). Generally, the AGO1 gene negatively affects the transcription of targeted miRNA genes (*Bajczyk et al., 2019*). In *A. thaliana*, AGO1 plays a crucial role in cotranscriptionally regulating the expression of MIR161 and MIR173 in response to

salinity stress (*Dolata et al., 2016*). Subsequent investigations revealed that overexpressing miR393 confers cold stress tolerance (*Liu et al., 2017*). It has been reported that miRNAs play a significant role as posttranscriptional regulators against drought, chilling stress, and *Colletotrichum gloeosporioides* attack in *Camellia sinensis* (*Guo et al., 2017*; *Jeyaraj et al., 2019*). In addition, miRNAs are involved in mediating resistance against iron deficiency in plants. For example, miRNA-mediated iron deficiency tolerance has been associated with increased stress resistance *via* reduced miR172 expression in *Citrus sinensis* (*Jin et al., 2021*). AGOs play a prominent role in regulating miRNA expression and plant growth.

## BIOLOGICAL ROLES OF PLANT AGOS IN ABIOTIC STRESS

### Drought stress

Drought is a significant environmental stress that adversely affects reproductive and vegetative plant growth and development by altering plant physiology, limiting metabolic pathways, and reducing crop productivity. It can also cause various biochemical and morphological alterations in plants (*Raza et al., 2023a*; *Raza et al., 2023b*). Many AGOs and miRNAs have been shown to play fundamental roles in regulating drought stress in various crop plants, such as *Solanum lycopersicum* (*Bai et al., 2012*), *A. thaliana* (*Yan et al., 2016*), *Populus trichocarpa* (*Zhao et al., 2015a*; *Zhao et al., 2015b*), *T. aestivum* (*Akdogan et al., 2016*) and *Oryza sativa* (*Zhang, Lang & Zhu, 2018*). Increased expression of *AtAGO1* and miR168a has been observed during exposure to drought stress in *A. thaliana*, playing a significant role in drought stress tolerance (*Li et al., 2012*). Genome-wide studies have shown that *SiAGO08* is markedly upregulated after 24 h of drought stress, whereas *SiAGO13* expression is increased after 1 h of drought stress in *Setaria italica* (*Yadav et al., 2015*). A mutation in the *SiAGO1b* gene leads to decreased resistance against drought stress in *S. italica* (*Liu et al., 2016*).

Downregulation of the *SlAGO4A* gene in transgenic tomato plants enhanced resistance to drought stress with greater membrane stability than did *SlAGO4A* overexpression in wild-type plants through the modulation of DNA methylation (*Huang et al., 2016*). Genome-wide analysis revealed that the expression of the *MdAGO1.1*, *4.1*, *5*, *6*, *7.2*, *9*, *10.2*, *12*, *13*, and *14* genes in *Malus domestica* was significantly increased in response to polyethylene glycol (PEG)-mediated drought stress (*Xu et al., 2016a*; *Xu et al., 2016b*). Among these *MdAGOs*, *MdAGO4.1* is significantly upregulated in response to drought and ABA stresses, demonstrating its essential role in both drought and ABA signaling pathways (*Zhou et al., 2016*). The expression of *CsAGOs* in cucumber (*C. sativa*) during abiotic stresses has been investigated, and *CsAGO1c*, *CsAGO6*, and *CsAGO10* were found to be significantly upregulated in roots under drought stress (*Gan et al., 2017*). Similarly, other *AGO* genes, namely, *CaAGO2* and *CaAGO10b*, in *Capsicum annuum* are involved in mediating resistance against drought stress in pepper. Furthermore, *CaAGO10b* regulates ABA-reactive genes and plays an essential role in protecting against osmotic stress in pepper (*Qin et al., 2018*). The expression patterns of *ZmAGOs* in response to drought stress were examined using RT−qPCR, and the *ZmAGO1*, *2b*, *4*, *5*, *7*, *9*, *10-b*, and *18a-b* genes were significantly upregulated after 1 h and subsequently downregulated after 2 and 4 h of

drought stress in maize. Furthermore, *ZmAGO18a* and *ZmAGO18b* were strongly induced after 1 h of dehydration, indicating their fundamental role in gene regulation in response to drought stress (*Zhai et al., 2019*). The role of *SsAGOs* in *Saccharum spontaneum* against dehydration stress was investigated. The expression profiles of 18 genes in young SES208 sugarcane plants exposed to PEG6000 were determined *via* qRT-PCR. *SsAGO2b*, *5a*, *5c*, *6b*, *10a*, and *10c* were significantly upregulated, while *SsAGO18b* was downregulated in response to PEG-mediated drought stress. Additionally, expression analysis revealed that *SsAGO2b*, *SsAGO5a*, *SsAGO6b*, and *SsAGO10c* exhibit enhanced expression in response to osmotic stress in sugarcane (*Cui et al., 2020*). According to recent suggestions, *TaAGO2A*, *5B*, *5D*, *6A*, *9*, and *17* have been shown to be significantly affected by PEG and salinity stresses, indicating their role in the resistance of wheat to various abiotic stresses (*Liu et al., 2021*). Analysis of the differential expression of *CsAGO* genes in tea (*C. sinensis*) under drought stress revealed that *CsAGO2/3a* and *CsAGO10c* were downregulated, while *CsAGO2/3f* was significantly upregulated (*Krishnatreya et al., 2021*). Furthermore, the expression of *Dimocarpus longan DlAGO1*, *DlAGO2*, *DlAGO4*, and *DlAGO10* was upregulated after 2 h of PEG-4000-mediated treatment and then gradually downregulated by 4 and 8 h, thereby conferring resistance against drought stress (*Chen et al., 2021*). AGOs play a key role in regulating drought stress resistance in various crop plants through signaling pathways.

## Salinity stress

Salinity stress negatively affects plant growth and development, causing metabolic, physiological, and biochemical alterations and leading to low crop yields worldwide (*Raza et al., 2022*). Approximately 20% of the world's irrigated land has been affected by salinity stress (*Kumar, 2020*). AGOs play an essential role in salinity stress resistance in plants. AtAGO1 binds to chromatin MIR161 or 173 complexes and releases non-polyadenylated transcripts by disassembling transcriptional precursors, thereby improving salinity stress resistance in *A. thaliana* (*Liu et al., 2018a*; *Liu et al., 2018b*). Genetic studies revealed that the positive actions of the *OsAGO1*, *2*, *4*, *16*, and *OsPNH1* genes in *O. sativa* were upregulated in response to salinity, dehydration, and cold stresses through microarray analysis (*Kapoor et al., 2008*). It has been reported that ring-type copine (McCPN1), in association with the AGO4 protein, facilitates salinity stress tolerance in ice plants (*Mesembryanthemum crystallinum*) (*Li et al., 2014*). Cellular abscisic acid (ABA) is associated with stress signaling pathways, salinity, and drought stress tolerance in plants (*Fernando & Schroeder, 2016*). The expression of *SiAGO10* and *SiAGO18* in *S. italica* significantly increased after 1 h of ABA stress (*Yadav et al., 2015*). In another study, increased tolerance to ABA stress was confirmed by the overexpression of *MdAGO4.1* in apple, indicating that the gene functions in ABA signaling (*Zhou et al., 2016*). The pea (*Pisum sativum*) p68 gene *Psp68* interacts with the AGO1 protein. In rice, the overexpression of *Psp68* confers enhanced tolerance to salinity stress (*Banu et al., 2015*). Moreover, *OsAGO2* positively affects salinity stress resistance by triggering *Big Grain 3* (*BG3*) expression in rice (*Yin et al., 2020*). In *A. thaliana*, AtAGO3 associates with 24-nucleotide sRNAs and plays a significant role in the RdDM pathway in response to high

salinity stress conditions (*Zhang et al., 2016*). In addition, *MdAGO3*, *4.1*, *5.2*, *6*, *7.2*, *8*, and *9* were significantly upregulated in response to 200 mM NaCl, demonstrating their role in the salinity stress response in *M. domestica* (*Zhou et al., 2016*). Most recently, *MdAGO1* has been shown to regulate ionic imbalance, scavenge ROS, and enhance polyamine accumulation in various plants under salinity stress. The expression of *MdAGO1* was significantly induced under salinity stress (*Wang et al., 2023*).

The constitutive expressions of *CaAGO4b*, *CaAGO7*, and *CaAGO10a* have been significantly suppressed in response to salinity stress. Moreover, *CaAGO1b*, *CaAGO2*, and *CaAGO5* have been shown to be induced in response to cold stress, indicating their potential role in resistance to salinity and cold stresses in *C. annuum* (*Qin et al., 2018*). In *A. thaliana*, MUG13.4, an RNA binding protein, plays an essential role in conferring salinity stress resistance by associating with AtAGO2. Furthermore, AtAGO2 improved salt stress tolerance by affecting the SOS signaling cascade. However, the role of AtAGO2 in preventing salinity stress is MUG13.4-dependent (*Wang et al., 2019*). When maize plants were exposed to NaCl, *ZmAGOs* were activated. *ZmAGOs* bind to sRNAs, which guide them to target specific mRNAs associated with the salinity stress response, enabling the repression of gene expression. In addition, *ZmAGO1a*, *5b-d*, *7*, and *10a* were slightly downregulated in response to salinity stress, while *ZmAGO1b*, 2b, 4, *5a*, and *18a* were slightly upregulated (*Zhai et al., 2019*). Interestingly, qRT-PCR revealed that *D. longan DlAGO1*, *2*, *4*, *5*, *10*, and *MEL-1* were upregulated but that *DiAGO6* was downregulated after 150 mmol NaCl treatment (*Chen et al., 2021*). Therefore, AGOs positively enhance salinity stress tolerance in plants by affecting the SOS signaling cascade.

## Freezing and chilling stress

Cold stresses detain plant development and often cause cellular death through antioxidant enzymatic defense activities, a reduction in membrane integrity, the biogenesis of secondary metabolites, and altered phytohormonal signaling (*Soualiou et al., 2022*). *SlAGO1a-b*, *4a-b*, and *5* in *Solanum lycopersicum* were shown to be highly expressed in response to heat, cold, and salinity stresses (*Bai et al., 2012*). In *A. thaliana*, *AtAGO1* has been reported to regulate cold and heat stresses through signaling pathways (*Liu et al., 2018a*; *Liu et al., 2018b*). The expression of *AGO1* is significantly downregulated in response to high and low temperatures in *Laodelphax striatellus*, whereas the expression of *AGO2* is affected by low temperature (*Zhou et al., 2016*). Genome-wide analysis revealed that the expression of *MdAGO1.2*, 1.3, 5.1, *7.2*, *10*, *14*, and *15* genes in *M. domestica* increased in response to cold stress at 4 °C, whereas the expression of *MdAGO1*, *3*, and *6* decreased (*Xu et al., 2016a*; *Xu et al., 2016b*; *Zhou et al., 2016*). In maize, *ZmAGOs* were shown to play a positive role in temperature fluctuations. After 1 h of cold stress, the expression of *ZmAGO1a/b/f*, *ZmAGO4*, and *ZmAGO9* was upregulated, and *ZmAGO5d* and *ZmAGO10a* were downregulated (*Zhai et al., 2019*). Additionally, when *Volvariella volvacea* encountered chilling stress, AGO1 was activated and bound to sRNAs to target and suppress the expression of genes associated with cold-induced damage. Interestingly, transcriptome data revealed that overexpression of *VvAGO1* in *V. volvacea* confers enhanced resistance against chilling stress by affecting ubiquitination or signal transduction pathways (*Gong et al., 2020*).

Overall, AGOs significantly improve resistance against low- and high-temperature stresses through signal transduction pathways.

## Heat stress

Temperature fluctuations, such as heat stress, cause adverse effects on the metabolic, enzymatic, physiological, growth, distribution, and developmental processes of plants (*Ding, Shi & Yang, 2020*; *Raza et al., 2023a*; *Raza et al., 2023b*). *SlAGO10a* and *SlAGO10b* have been shown to be expressed explicitly in response to heat stress (*Bai et al., 2012*). In *M. domestica*, *MdAGO2* is overexpressed under heat stress and acts as an RNA silencing effector, facilitating the posttranscriptional regulation of target genes involved in heat stress responses (*Zhou et al., 2016*). The expression of *ZmAGO1f*, *2b*, *4*, *5a-b-c*, *9*, and *10b* is upregulated after 1 h of treatment with 40 °C heat stress in maize (*Zhai et al., 2019*). In wheat, *TaAGO5B* and *TaAGO6B* are significantly upregulated after 6 h and 24 h of heat stress, whereas *TaAGO5A*, *TaAGO6A*, and *TaAGO6D* are downregulated after 6 h and then upregulated after 24 h of heat stress (*Liu et al., 2021*).

## Other abiotic stresses

UV-B radiation affects leaf thickness, development, and growth of crop plants (*Thomas, Dinakar & Puthur, 2020*). AGO1 participates in the repair of UV-induced DNA damage by binding UV-induced RNAs (uviRNAs) and forming a complex with DNA damage binding protein 2 (DDB2) (*Bajczyk et al., 2019*). Heavy metals enhance the production of ROS, which oxidatively damage plants (*Talukder et al., 2023*). It has been reported that miR168 targets *AGO1* and plays a significant role in mediating resistance against cadmium (Cd) stress (*Ding, Chen & Zhu, 2011*). Similarly, downregulating miR162 and miR168 while upregulating *AGO1* and *DCL* transcripts may posttranscriptionally regulate gene expression in response to arsenic (As) stress in maize (*Ghosh et al., 2022*). Recent studies have indicated that the *AtAGO1* and *AtAGO4-* dependent RdDM pathways are also involved in the regulation of hypoxia tolerance in *A. thaliana* (*Loreti et al., 2020*). AGO genes play a crucial role in plant cell signaling and confer enhanced tolerance against different abiotic stresses in plants.

## BIOLOGICAL ROLES OF PLANT AGOS IN BIOTIC STRESS

### Fungal stress

AGO proteins are reportedly associated with fungal stress resistance in plants (*Nguyen et al., 2018*). The AGO1 gene plays a significant role in the response to fungal stress. In *Botrytis cinerea*, sRNAs associated with *A. thaliana* AGO1 hijacked the RNAi pathway, thus silencing host resistance genes. In response, the *A. thaliana* AGO1 mutant exhibited decreased susceptibility to *B. cinerea* infection (*Weiberg et al., 2013*). In *A. thaliana*, the *AtAGO1*, *2*, *7*, and *9* genes exhibited increased susceptibility to *Phytophthora* species and *Sclerotinia sclerotiorum*. These AGOs regulate the expression of defense-related genes through PTGS activity, consequently altering resistance against fungal pathogens (*Cao et al., 2016a*; *Guo et al., 2018*). AGOs and miRNAs were shown to regulate stress resistance in various plants in response to fungal pathogens such as *Mucor circinelloides* (*Cervantes*

*et al., 2013*), *Verticillium longisporum* (*Shen et al., 2014*), and *Colletotrichum higginsianum* (*Campo, Gilbert & Carrington, 2016*). The *BnAGO2a*, *2b*, *3a*, *3b*, and *5a* genes in *Brassica napus* are significantly induced during *Sclerotinia sclerotiorum* infection (*Zhao et al., 2016*). Transcriptome analysis revealed that *VMAGO2* is significantly upregulated in response to *Valsa mali* infection, indicating that *VMAGO2* is involved in fungal pathogenicity and $H_2O_2$ tolerance (*Feng et al., 2017*). In addition, *CaAGO7* in *Cicer arietinum* was shown to be involved in mediating antifungal defense against *Ascochyta blight* infection. Overexpression of the *CaAGO4* gene confers resistance in response to *Ascochyta rabiei* fungal infection (*Garg et al., 2017*). Furthermore, when *Nicotiana attenuata* was infected with *Fusarium brachygibbosum*, the expression of *NaAGO4* was induced. *NaAGO4* can bind to sRNAs that are generated in response to fungal attack. These siRNAs guide *NaAGO4* to target mRNAs from the pathogen *via* complementary sequences. By binding to these mRNAs, *NaAGO4* triggers posttranscriptional gene silencing by affecting the expression of genes involved in the plant defense response. These results demonstrated that *NaAGO4* is involved in maintaining resistance against *F. brachygibbosum* (a hemibiotrophic pathogen) by regulating jasmonic acid (JA) biogenesis and JA signaling (*Pradhan et al., 2020*). Moreover, another defense gene, *StAGO15*, positively regulates stress resistance to *Phytophthora infestans* in *Solanum tuberosum* (*Liao et al., 2020*).

Increased expression of *AGO2* and *AGO4* was shown to be associated with defense against cowpea mosaic virus (CPSMV) (*Martins et al., 2020*). Leaf stripe is a major disease caused by *Pyrenophora graminea* that reduces barley quality and yield (*Si et al., 2020*). *HvAGO1*, *HvAGO2*, and *HvAGO4* are significantly increased in *Pyrenophora graminea* infection (*Yao et al., 2021*). Similarly, the downregulation of *OsAGO1* in rice increased susceptibility to *Magnaporthe oryzae*. Subsequently, *OsAGO1* has been associated with miR168 and plays an essential role in improving yield, reducing flowering time, and mediating resistance against *M. oryzae* in rice (*Wang et al., 2021a*; *Wang et al., 2021b*). Furthermore, the upregulation of *PvAGO2a*, *PvAGO4a*, and *PvAGO4b* in *Phaseolus vulgaris* enhanced antifungal stress resistance during *Colletotrichum lindemuthianum* infection (*Alvarez-Diaz et al., 2022*). Expression analysis revealed that the *GhAGO2a*, *4a*, *7c*, and *7d* genes in cotton were significantly upregulated in response to *Verticillium dahliae* infection, while the *GhAGO1b*, *2d*, *5b*, *5d*, *7a*, *7b*, and *10b* genes were downregulated (*Fu et al., 2022*). Most recently, VaAGOs were shown to exhibit different expression patterns against BCMV and *Podosphaera xanthii* infection in adzuki bean (*Vigna angularis*), and *VaAGO1*, *3a*, *3b*, *4b*, *6*, and *7b* were downregulated for defense against *Podosphaera xanthii* (*Li et al., 2023*). Overall, AGOs are involved in plant development and the regulation of fungal defense through signaling pathways.

## Bacterial stress

Various AGOs play crucial roles in preventing bacterial infection in different crop plants (*Fátyol, Ludman & Burgyán, 2016*). In *A. thaliana*, AtAGO2 associates with miR393b* and silences the *MEMB12* gene by translational suppression, thus increasing the exocytosis of the pathogenesis-related protein PR1 and antibacterial defense responses (*Zhang et al., 2011*). Various abiotic and biotic stresses promote the constitutive expression of the

AGO2 gene, which plays a significant role in preventing stresses from ionizing radiation and bacterial and viral infection (*Wei et al., 2012*). Moreover, AGO5 was shown to be involved in resistance against rhizobial infection in *Glycine max* and *P. vulgaris* (*Reyero-Saavedra et al., 2017*). *AtAGO1* and *AtAGO2* enhanced antibacterial and antiviral resistance in *A. thaliana* (*Ludman, Burgyán & Fátyol, 2017*). Recently, it has been shown that reduced arginine methylation of AGO2 caused by decreased expression of protein arginine methyltransferase (PRMT5) contributes to the accumulation of AGO2- and AGO2-associated sRNAs *via* reduced interactions between AGO2 and Tudor domain proteins (TSNs), which promote plant resistance (*Hu et al., 2019*; *Zhang et al., 2019*). Sugarcane plants suffer from leaf scald bacterial disease caused by *Xanthomonas albilineans,* which reduces the juice quality and cane yield in China (*Ntambo et al., 2019*). When sugarcane SES208 plants were infected with *X. albilineans*, *SsAGOs* bound to the sRNAs were produced in response to the infection. These sRNAs guided *SsAGO6B* and *SsAGO10c* to target specific mRNAs from *X. albilineans*. By binding to these mRNAs, AGO proteins promote posttranscriptional gene silencing, effectively disrupting the expression of essential bacterial genes, which impedes pathogen growth and virulence. The transcript expression patterns of *SsAGOs* were analyzed using RT-qPCR in *S. spontaneum*. *SsAGO6b* and *SsAGO10c* are significantly upregulated, while *SsAGO10e*, *SsAGO18b*, and *SsAGO18c* are downregulated after *X. albilineans* infection (*Cui et al., 2020*). AGOs exhibit different expression patterns in plants and play significant roles in antibacterial stress resistance. AGO-mediated signaling cascades and defense responses play essential roles in enhancing resistance to pathogenesis caused by different biotic stresses.

## Viral stress

Plant viruses have a negative impact on sustainable agriculture worldwide (*Rubio, Galipienso & Ferriol, 2020*). AGOs participate in the degradation of viral RNA, indicating their role in plant antiviral stress resistance (*Musidlak, Nawrot & Goździcka-Józefiak, 2017*). These proteins have been shown to regulate antiviral resistance by overexpressing AGO1 and AGO2, which bind to viral siRNAs and form antiviral RNA silencing complexes (*Garcia-Ruiz et al., 2015*). Various *AGO* genes, such as *AGO1*, *2*, *4*, *5*, *7*, and *10* in *A. thaliana*, *AGO1* and *18* in *O. sativa,* and *AGO1* and *2* from *N. benthamiana*, have been found in plants and play prominent roles in antiviral stress resistance (*Carbonell & Carrington, 2015*; *Alazem et al., 2017*). In addition, *AtAGO1* was observed to be upregulated during viral infection, during which vsiRNAs interact with AGO1 to enhance viral RNA degradation (*Guo, Li & Ding, 2019*). Similarly, overexpression of *AtAGO1* confers resistance against turnip crinkle virus (TCV) and brome mosaic virus (BMV) (*Dzianott, Sztuba-Solińska & Bujarski, 2012*; *Zhang et al., 2012*). When turnip mosaic virus (TuMV) infects a plant, it triggers the production of virus-derived siRNAs. AGOs are associated with these siRNAs and guide the protein to target complementary sequences on the viral RNA. AGO2 cleaves viral RNA and silences the TuMV genes, revealing its essential role in antiviral resistance in response to TuMV (*Garcia-Ruiz et al., 2015*). Genetic studies have shown that *AtAGO4* is involved in antiviral stress resistance in response to *Plantago asiatica* mosaic virus (*Brosseau et al., 2016*).
Biochemical and genetic evidence revealed that overexpression of AGO1 and AGO18 in rice enhanced antiviral stress resistance against rice dwarf phytoreovirus and rice stripe tenuivirus. AGO18 binds to miR168 and plays a significant role in the growth and development of plants and in viral defense. Similarly, AGO18 sequesters miR168 from AGO1 in rice and regulates antiviral resistance against rice stripe virus (RSV) and rice dwarf virus (RDV) (*Wu et al., 2015*). Furthermore, AGO18 sequesters miR528 *via* AGO1 to control antiviral resistance. It inhibits the cleavage of ascorbic acid oxidase (AO) mRNA from AGO1-linked miR528 and enhances the AO-mediated acquisition of ROS, thereby increasing antiviral resistance in rice (*Wu et al., 2017*; *Yang et al., 2020*). It has been suggested that osa-siRNAs produced by *OsDCL4* in rice regulate their target gene expression through the *OsAGO18* pathway (*Niu et al., 2018*). Moreover, miR444 has been associated with *OsAGO1*, thus alleviating translational suppression of the *OsRDR1* gene and promoting antiviral stress resistance against rice stripe virus infection (*Wang et al., 2016*). Additionally, *OsAGO2* plays a significant role in controlling the expression of *HEXOKINASE1* to modulate ROS accumulation in response to rice black streak dwarf virus (*Wang et al., 2021a*). In *N. benthamiana*, *AGO2* has been demonstrated to play an essential role against potato virus X (*Fátyol, Ludman & Burgyán, 2016*), tomato ringspot virus (*Paudel et al., 2018*), tobamovirus (TMV) (*Diao et al., 2019*), tombusvirus (TCV) and potyvirus (TuMV) (*Ludman, Burgyán & Fátyol, 2017*). *AGO1* is induced in the RNA silencing pathway and was shown to be involved in resistance against tomato ringspot virus (*Ghoshal & Sanfaçon, 2014*), tombusvirus (*Kontra et al., 2016*), and tobacco rattle virus (*Odokonyero et al., 2017*). *AGO1* and *AGO10* positively regulate defense mechanisms in response to bamboo mosaic virus. Knockdown of *AGO1* enhanced BaMV accumulation, while knockdown of *AGO10* negatively regulated BaMV accumulation in *N. benthamiana* (*Huang et al., 2019a*; *Huang et al., 2019b*). When pepper leaves were inoculated with tobacco mosaic virus (TMV), potato virus Y (PVY), or cucumber mosaic virus (CMV), *CaAGOs* associated with the viral siRNAs were found to cleave the viral RNA and effectively silence the viral genes. qRT-PCR analysis revealed that the expression of the *CaAGO2*, *6*, and *10b* genes in *C. annuum* was significantly upregulated in response to TMV, PVY, and CMV infection (*Qin et al., 2018*). Additionally, image-derived phenotyping approaches have shown that *AGO2* and *AGO7* regulate antiviral resistance against turnip crinkle virus infection (*Zheng et al., 2019*). Similarly, *TaAGO5* was shown to enhance antiviral resistance in *T. aestivum* (*Sibisi & Venter, 2020*). PaAGO5a, PaAGO5b, and PaAGO10s play crucial roles in antiviral resistance against Odontoglossum ringspot virus (ORSV) and Cymbidium mosaic virus (CymMV) in *Phalaenopsis aphrodite* (*Kuo et al., 2021*). More recently, it was reported that knocking out or overexpressing *NtAGO1* in *N. tabacum* by *Agrobacterium*-mediated transformation confers antiviral resistance (*Han et al., 2022*). Overexpression of AGOs confers enhanced antiviral stress tolerance in various plants.

## Insect stress

miRNAs are involved in the regulatory pathways of plants subjected to insect infestation by overexpressing or downregulating different genes associated with insect-plant interactions (*Ražná & Cagáň, 2019*; *Bordoloi & Agarwala, 2021*). Previously, miR167 and miR393 were

shown to activate and develop auxin insensitivity against aphid infestation in *Cucumis melo* (*Sattar, Addo-Quaye & Thompson, 2016*). In sweet potato, miR408 overexpression reduced resistance to *Spodoptera litura*, suggesting that miR408 plays a defensive role against insect stress (*Kuo et al., 2019*). AGOs play a significant role in regulating insect stress resistance *via* miRNAs. For example, *AGO2* is involved in resistance against *Manduca sexta* (*Garbutt & Reynolds, 2012*) and *Acyrthosiphon pisum* (*Ye et al., 2019*). In addition, *NaAGO8* in *N. attenuata* is associated with miRNAs and confers resistance against herbivory attack by regulating several metabolites in a defense signaling network (*Pradhan et al., 2017*). Furthermore, wheat plants can detect aphid infestation and initiate sRNA-mediated pathways. *TaAGO5* proteins bind to aphid-induced sRNAs and guide them to target mRNAs with complementary sequences. This leads to posttranscriptional gene silencing, which inhibits the expression of genes involved in herbivore resistance, defense mechanisms, and signaling pathways. The downregulation of *TaAGO5* in wheat confers resistance against *Diuraphis noxia* infection and other insects (*Sibisi & Venter, 2020*).

## CONCLUSION

This review has summarized the updated molecular features and biological functions of AGOs in different plants over the last two decades. Plant AGOs are involved in gene regulation through chromatin modification, DNA methylation, RNA slicing, and translational suppression, thus influencing plant development and metabolic processes that may be associated with responses to various biotic and abiotic stresses (Table 2). These findings might be helpful for using RNAi pathways in crop breeding for stress resistance. Furthermore, exploring the functions and regulatory mechanisms of AGO proteins provides a deeper understanding of these proteins in controlling plant immunity, growth, and development. This knowledge will lay the foundation for developing innovative strategies for engineering stress-resistant crops, creating novel biotechnological applications, and advancing sustainable agriculture.

### Funding
The work was supported by the National Natural Science Foundation of China (32172503) and the Natural Science Foundation of Fujian Province (2023J01069). The funders had no role in study design, data collection and analysis, decision to publish, or preparation of the manuscript.

### Grant Disclosures
The following grant information was disclosed by the authors:
National Natural Science Foundation of China: 32172503.
Natural Science Foundation of Fujian Province: 2023J01069.

### Competing Interests
The authors proclaim no conflict of interest.

**Table 2   Functions of ARGONAUTEs in plant responses to stresses.**

| Plant AGOs | Mode of action | References |
|---|---|---|
| AtAGO1 | AtAGO1 interacts with MIR161/173 complexes and releases un-polyadenylated transcripts to improve salinity stress tolerance | *Liu et al. (2018a)*; *Liu et al. (2018b)* |
| VvAGO1 | Over-expression of VvAGO1 conferred resistance against chilling stress through affecting ubiquitination | *Gong et al. (2020)* |
| AtAGO2 | AtAGO2 enhanced salinity stress tolerance by affecting SOS signaling cascade | *Wang et al. (2019)* |
| OsAGO2 | OsAGO2 played a significant role in salt tolerance by triggering Big Grain 3 expression in rice | *Yin et al. (2020)* |
| SlAGO4A | SlAGO4 has been shown to confer drought stress resistance by modulating DNA methylation | *Huang et al. (2016)* |
| NaAGO4 | Ectopic expressions of NaAGO4 conferred resistance against *Fusarium brachygibbosum* by regulating JA biogenesis and signaling | *Pradhan et al. (2020)* |
| CaAGO10b | CaAGO10b regulated ABA reactive genes which in turn enhance the osmotic stress resistance | *Qin et al. (2018)* |
| TaAGO6B | TaAGO6B was up-regulated and conferred resistance to heat stress | *Liu et al. (2021)* |
| CaAGO7 | CaAGO7 was shown to be involved in mediating antifungal stress resistance against *Ascochyta blight* infection | *Garg et al. (2017)* |
| NaAGO8 | NaAGO8 conferred resistance under herbivory attack by regulating several metabolites in a defense signaling network | *Pradhan et al. (2017)* |
| TaAGO5 | Down-regulation of TaAGO5 conferred resistance against *Diuraphis noxia* infestation | *Sibisi & Venter (2020)* |
| SsAGO10c | Expressions of SsAGO10c was significantly up-regulated after *Xanthomonas albilineans* infection | *Cui et al. (2020)* |
| StAGO15 | StAGO15 positively regulated stress resistance to *Phytophthora infestans* | *Liao et al. (2020)* |
| OsAGO18 | OsAGO18 sequestered miR168 from AGO1 and regulated antiviral defense against rice stripe virus and rice dwarf virus | *Wu et al. (2015)* |

**Notes.**

At, *Arabidopsis thaliana*; Vv, *Volvariella volvacea*; Os, *Oryza sativa*; Sl, *Solanum lycopersicum*; Na, *Nicotiana attenuata*; Ca, *Capsicum annuum*; Ta, *Triticum aestivum*; Ss, *Saccharum spontaneum*; St, *Solanum tuberosum*.

## Author Contributions

- Uroosa Zaheer conceived and designed the experiments, performed the experiments, prepared figures and/or tables, authored or reviewed drafts of the article, and approved the final draft.
- Faisal Munir analyzed the data, prepared figures and/or tables, and approved the final draft.
- Yussuf Mohamed Salum analyzed the data, prepared figures and/or tables, and approved the final draft.
- Weiyi He conceived and designed the experiments, authored or reviewed drafts of the article, and approved the final draft.

## Data Availability

The raw data for domain analysis, bootstrap values for tree, AGO18 protein sequences are available in the Supplemental Files.

## Supplemental Information

Supplemental information for this article can be found online at http://dx.doi.org/10.7717/peerj.17115#supplemental-information.

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
