# Peer review of "Function and regulation of plant ARGONAUTE proteins in response to environmental challenges: a review"

_PeerJ, doi:10.7717/peerj.17115_

## Round 0.1 · original submission · Major Revisions

Please take into account all the reviewer{s comments, in particular, proper citation of key references in the area (Matzke, M. A., & Mosher, R. A. (2014). RNA-directed DNA methylation: an epigenetic pathway of increasing complexity. Nature Reviews Genetics, 15(6), 394-408.)

Reviewer 1 ·

Basic reporting

Here the authors make a compilation of the functions of AGO in plants under different types of stress. The information is relevant and interesting, but there are some points that need to be improved.

Minor comments:
Lines 47-51: Climate change is part of abiotic stress; it can be included in that category.

Lines 93-95: If you want to compare the lesser and the highest number of AGO in plants, there are plants that possess fewer of these proteins in their genomes than Cucumis sativus, such as Arabidopsis. Furthermore, considering that most of the information regarding these topics has been explored in this model plant, it is necessary to include it in your table 1.

Line 162-164: Are you talking about AtAGO1 and AtAGO10 genes or proteins?

Lines 208-209: These components are essentials for RdDM not only in O. sativa, but in all plants.

Lines 226-231: This is repetitive information.

It is not necessary that you write the complete scientific names of plants every time. With the first time is enough.

Line 420: AtAGO2 must be written in flat letters meanwhile MEMB12 in italics. Please check that all your proteins are written in plain letters and all your genes in italics.


Major comments:

Figure 1. Which sequence did you use to model the AGO structure?

Lines 136-138: What do you mean by AGO2 with AGO1? Do you mean that these proteins interact between them to silencing pseudogenes, transposons, and intergenic regions?

In general, you give many examples; however, you don’t deep into the molecular mechanisms of any of them. It would be interesting if you described at least one complete mechanism for each type of stress.

Line 349: What do you mean with “optimistic role”?

Lines 383-385: Weiberg et al., 2013, demonstrated that B. cinerea sRNAs hijack the RNAi silencing machinery of Arabidopsis, including AGO1 to selectively silence host immunity genes. Thus, how it’s possible that you say that over-expression of AGO1 exhibits enhanced tolerance against this phytopathogen? You need to read carefully all the papers you are citing in your manuscript. In this specific case, you are stating the opposite of what the authors discovered.

Lines 385-387: How does a gene exhibit vulnerability against a microorganism? Are you talking about their expression levels?

Instead of your Figure 4, I think that a table with a summary of the function of AGOs during different stresses cited here could be more helpful.

Experimental design

You have a lot of papers cited throughout your manuscript. It strikes me that some of the basic works on these topics are only cited once (i.e. Borges et al., 2015) and some others are missing (i.e. Matzke, M. A., & Mosher, R. A. (2014). RNA-directed DNA methylation: an epigenetic pathway of increasing complexity. Nature Reviews Genetics, 15(6), 394-408.) I suggest you carefully review the works you are citing and leave only the most relevant ones.

Validity of the findings

The conclusion could be improved in order to reflect how this information could help in the study field.

Additional comments

No comments

Reviewer 2 ·

Basic reporting

The manuscript provides a comprehensive overview of the biological roles of plant Argonaute (AGO) proteins in response to various abiotic and biotic stresses, including drought, salinity, temperature stresses (freezing, chilling, and heat), as well as fungal, bacterial, viral, and insect stresses.

Experimental design

Please provide
1. explanation of the miRNA biogenesis pathway to provide a more complete context for readers less familiar with miRNA processing.
2. some insights into how AGO-miRNA complexes are loaded onto AGOs and the factors influencing this process would enhance the understanding of AGO-miRNA interactions.
3. Further elaboration on the role of AGO proteins in stress adaptation and their implications in enhancing plant resilience to environmental stress would add depth to the discussion.

Validity of the findings

Conclusion is very short

Additional comments

Elaborate the Conclusions

---

## Round 0.2 · Minor Revisions

Could you please look at the reviewer's comments and address them? To get their feedback incorporated into the rmanuscript. Thanks!

Reviewer 1 ·

Basic reporting

The authors improved their manuscript. Also, they clarified some confusing points. I have a few points:

-Please carefully review the nomenclature of genes and proteins. Some genes are written in lowercase and others in uppercase. Also, some are not written in italics (i.e. line 492: NaAGO4).

You have some grammatical mistakes, i.e.:
259: hairpin
263: afterward
266: resulting
267: remove the parenthesis before "Furthermore"
671: remove the parenthesis

Your conclusion is repetitive, please consider improving it.

Experimental design

No comment

Validity of the findings

No comment

Additional comments

No comment

---

## Round 0.3 · Minor Revisions

Before resubmitting the manuscript, please provide evidence of professional proofreading to address the many grammar and style errors that still need to be fixed, despite the revisions made.

**Language Note:** The Academic Editor has identified that the English language must be improved. PeerJ can provide language editing services - please contact us at copyediting@peerj.com for pricing (be sure to provide your manuscript number and title). Alternatively, you should make your own arrangements to improve the language quality and provide details in your response letter. – PeerJ Staff

---

## Round 0.4 · Minor Revisions

I regretfully notice that the changes requested as "minor revisions" have not been addressed with respect to improving the English language used throughout.

I request that for further publishing, authors use the services of a professional copyediting service.

---

## Round 0.5 · Minor Revisions

Please also proofread the Abstract.

**Language Note:** The Academic Editor has identified that the English language must be improved. PeerJ can provide language editing services - please contact us at copyediting@peerj.com for pricing (be sure to provide your manuscript number and title). Alternatively, you should make your own arrangements to improve the language quality and provide details in your response letter. – PeerJ Staff

---

## Round 0.6 · accepted · Accept

Thanks for fixing the abstract, your contribution is now accepted in PeerJ.